# Incidental Findings on Abdominopelvic CT in Young Korean Soldiers: Prevalence, Clinical Relevance, and Healthcare System Implications

**DOI:** 10.3390/healthcare13212736

**Published:** 2025-10-29

**Authors:** Kyungwon Lee, Kyung Uk Jung, Changsin Lee, Donghyoun Lee

**Affiliations:** 1Department of Critical Care Medicine, Uijeongbu Eulji Medical Center, Eulji University College of Medicine, Uijeongbu 11759, Republic of Korea; enmma23@eulji.ac.kr; 2Department of Surgery, Kangbuk Samsung Hospital, Sungkyunkwan University School of Medicine, Seoul 03181, Republic of Korea; sahel.jung@samsung.com; 3Department of Surgery, Armed Forces Capital Hospital, Seongnam 13574, Republic of Korea; viano0420@naver.com; 4Department of Surgery, Jeju National University Hospital, Jeju National University College of Medicine, Jeju 63241, Republic of Korea

**Keywords:** incidental findings, abdominopelvic CT, military healthcare, follow-up systems, health service readiness

## Abstract

**Background:** This retrospective case series examines incidental findings (IFs) detected on abdominopelvic CT (APCT) among young Korean soldiers presenting with acute abdominal pain. APCT is a frontline test for acute abdominal pain but frequently reveals incidental findings (IFs) unrelated to the presenting complaint. While many IFs are benign, some require structured follow-up. In military settings with constrained access and frequent personnel transfers, IFs pose challenges for health-system readiness. **Methods:** We retrospectively reviewed 1062 male Korean soldiers (18–28 years) who underwent APCT for acute abdominal pain at a military emergency department (ED) between January 2021 and December 2022. Two board-certified radiologists independently reassessed all scans to identify IFs and to classify those requiring follow-up based on contemporary guidelines. **Results:** IFs were identified in 218/1062 (20.5%) patients. Common categories included renal cysts (6.2%) and hepatobiliary IFs (7.5%). Clinically significant lesions comprised Bosniak IIF renal cysts (0.3%), inherited cystic kidney disease (0.2%), IPMN (0.1%), adrenal incidentalomas (0.4%), and appendiceal mucoceles (0.2%). An exploratory analysis suggested co-occurrence clusters (e.g., renal and hepatic cysts). **Conclusions:** IFs on APCT are prevalent even in a young, ostensibly healthy military cohort, highlighting a gap between detection and effective follow-up. Implementing structured reporting, automated tracking, and cross-institution referral pathways may mitigate long-term risk and support operational readiness in settings with limited subspecialty access and frequent relocations.

## 1. Introduction

Computed tomography has become a frontline diagnostic tool in emergency departments, particularly for patients with acute abdominal pain. The detailed anatomical imaging provided by abdominopelvic CT (APCT) facilitates accurate diagnosis of acute conditions. However, the comprehensive scope of APCT also leads to the identification of incidental findings (IFs)—abnormalities unrelated to the patient’s presenting symptoms. Importantly, IFs encompass any such unrelated abnormality regardless of clinical significance; while many IFs are benign or clinically unimportant, a subset may represent significant pathology requiring intervention or follow-up. Prior studies estimate that approximately 20–40% of emergency CT examinations reveal at least one incidental finding, with some reports of abdominal CTs in Emergency Department (ED) patients showing IF rates up to 34–43% and even as high as 56.3% in certain populations [1,2,3]. These findings underscore the need for a systematic approach to reviewing all APCT images to recognize potentially significant incidental abnormalities and ensure appropriate follow-up.

Despite their potential clinical importance, IFs are often underreported or inadequately managed in the ER setting. Emergency physicians must prioritize acute, symptomatic conditions, and time constraints can lead to incidental findings being overlooked or deferred. Prior research indicates that only about 20–25% of ED patients with incidental findings receive proper documentation and arranged outpatient follow-up, highlighting the risk of missed diagnoses [3,4,5]. Military hospitals may face additional challenges: limited availability of specialist radiologists, high patient turnover, and frequent relocations of personnel can all contribute to IFs being missed or not conveyed effectively. In South Korea, mandatory military service places young adult males in environments where access to comprehensive medical facilities is limited. A recent survey by Bae et al. found that nearly one in four Korean soldiers (24.8%) reported unmet healthcare needs during military service, a rate 6.8 times higher than among civilian males in their twenties (3.65%) [6]. Such limitations in military healthcare delivery underscore the importance of promptly identifying and clearly communicating IFs to prevent adverse outcomes.

This unique population is presumed healthy due to rigorous pre-enlistment screening, yet they operate under unique physical and environmental stressors [7]. Therefore, understanding the prevalence of underlying pathologies is paramount for maintaining force readiness. To our knowledge, this is the first study to characterize the spectrum of APCT incidental findings in this critical population [8].

## 2. Materials and Methods

### 2.1. Study Design and Population

Study design and population. We performed a retrospective case series of Korean soldiers who presented to the ED of our military hospital with acute abdominal pain between January 2021 and December 2022. Our hospital is the largest of 17 military hospitals in South Korea (666 beds, 130 physicians excluding trainees). The Institutional Review Board (IRB No. AFCH 2023-03-006) approved the study and waived informed consent due to the retrospective design. This cohort comprised all eligible encounters from our center and represents a convenience sample.

Eligibility. Inclusion criteria were: (i) male Korean soldiers aged 18–28 years; (ii) ED presentation primarily for acute abdominal pain; (iii) a contrast-enhanced APCT performed at our hospital during the index ED visit (intravenous iodinated contrast, portal venous phase); and (iv) complete, accessible medical records and imaging for review. Patients were excluded if: (a) they had undergone APCT at another facility prior to the index ED visit; (b) medical records or imaging were incomplete, unclear, or insufficient for accurate interpretation; or (c) they were transferred from external hospitals with known abdominal conditions that could confound identification of new incidental findings.

Cohort assembly and exclusions. Figure 1 illustrates the STROBE flow: 1610 encounters initially screened, 548 exclusions (predominantly missing or non-reviewable imaging and incomplete charts), and a final analytic cohort of 1062. Because the exclusion rate was high (34%), we summarize exclusion reasons to inform potential selection bias. Notably, the frequent exclusion of cases with prior outside APCT means our analytic cohort preferentially includes first-time imaging at our site, which could systematically influence IF prevalence estimates; findings should therefore be interpreted with caution.

### 2.2. Definitions and Classification of Incidental Findings

Incidental findings (IFs) were defined a priori as any radiologic abnormality on APCT that was unrelated to the presenting abdominal symptoms or the initial clinical diagnosis, regardless of clinical relevance. 

Clinically significant IFs were prespecified as lesions warranting surveillance, subspecialty referral, or intervention according to contemporary consensus statements and guidelines (Bosniak 2019; ESE/ENSAT; Fukuoka/Kyoto; recent hepatology guidance; Fleischner Society) [9,10,11,12,13,14,15]. Operational thresholds were as follows: renal cysts ≥ Bosniak IIF; adrenal lesions ≥ 1 cm prompting dedicated hormonal work-up; IPMN with worrisome features or high-risk stigmata; complex hepatic cyst features; and pulmonary nodules based on size and morphology per Fleischner criteria.

### 2.3. CT Acquisition and Protocol

All APCT examinations included in this analysis were acquired on multidetector CT scanners (Canon computed tomography TSX-305A; Canon Medical Systems Corporation, Otawara, Japan) with intravenous iodinated contrast in the portal venous phase. Images were reconstructed according to institutional standards with axial and multiplanar reconstructions.

### 2.4. Radiologic Assessment

All APCT examinations were initially interpreted by board-certified radiologists at the time of care. For this study, two senior radiologist consultants independently re-reviewed each scan in a blinded fashion (mutually blinded to each other’s assessments and to downstream outcomes; only the clinical indication was available). Discrepancies were resolved by consensus; if consensus could not be reached, a third senior radiologist adjudicated. Using the definitions in Section 2.2, we recorded the presence of any IF and whether it met criteria for clinical significance. Formal inter-observer agreement statistics (e.g., Cohen’s κ) were not calculated.

### 2.5. Statistical Analysis

All statistical analyses were performed using IBM SPSS Statistics, version 28.0 (IBM Corp., Armonk, NY, USA). Patient characteristics and IF frequencies were summarized with descriptive statistics. Age distribution was assessed for normality (Shapiro–Wilk test and visual inspection of histograms); because age was non-normal, continuous variables are reported as medians with interquartile ranges (IQR). Group differences in continuous variables were evaluated with the Mann–Whitney U test, and differences in categorical variables with chi-square tests; when any expected cell count was <5, Fisher’s exact test was used. Two-sided *p* < 0.05 was considered statistically significant for these general comparisons.

For descriptive visualization of pairwise co-occurrence among IF categories, we constructed an *m* × *m* count matrix (zeros indicating no co-occurrence) and rendered a symmetric heatmap with a linear color scale and in-cell counts; the display is presented in the Results section for context.

We assessed pairwise associations using chi-square or Fisher’s exact tests and reported ORs (95% CI). Multiple testing was controlled with Benjamini–Hochberg FDR, with significance at *q* < 0.05. Significant pairs are displayed in a network graph. Given the exploratory nature, cells < 5 were treated as hypothesis-generating.

Artificial intelligence software ChatGPT (GPT-5 Thinking) by OpenAI (San Francisco, CA, USA) was used only for language editing and formatting; no part of the scientific analysis or interpretation was generated by AI.

## 3. Results

### 3.1. Patient Inclusion and Characteristics

A total of 1062 young male soldiers with acute abdominal pain met the inclusion criteria and underwent APCT at the ED during the two-year study period. This final sample represents all eligible cases from our center (convenience sample). IFs—defined as radiological abnormalities unrelated to the initial abdominal complaint (irrespective of clinical importance)—were identified in 218 of these 1062 patients, corresponding to an overall incidental finding prevalence of 20.5%. The remaining 844 patients (79.5%) had no incidental findings on their APCT. The age distribution of the cohort was tightly clustered (median age 21 years, IQR 20–22) due to the narrow age range of conscripted soldiers in Korea. Patients with IFs had a median age of 21 (IQR 20–22), which did not differ significantly from those without IFs (median 21, IQR 20–22; *p* = 0.14). All patients were male by study design, and other baseline clinical characteristics did not significantly differ between those with and without IFs. The demographic and baseline characteristics of the study population are summarized in Table 1.

#### 3.1.1. Incidental Findings on APCT: Frequencies and Types

The anatomical distribution and frequency of incidental findings on APCT are detailed in Table 2. Renal-system IFs were most prevalent. Renal cysts were the single most common incidental finding, present in 66 patients (6.2% of the entire cohort) [1,13]. These cysts were predominantly simple benign cysts (Bosniak class I in 56 cases and class II in 5 cases). Notably, three renal cysts (0.3% of the cohort) were classified as Bosniak IIF, which are indeterminate and carry a potential malignancy risk, prompting a recommendation for scheduled radiologic follow-up [14]. In addition to simple cysts, we incidentally identified two significant renal conditions: one case of autosomal dominant polycystic kidney disease (ADPKD) and one case of medullary cystic kidney disease (MCKD). These were newly discovered in our young population (0.1% each of the total sample) and were clinically important, leading to referrals to specialized civilian centers for further management. Asymptomatic renal calculi (kidney stones) were also relatively frequent incidental findings, occurring in 20 patients (1.9%). While these stones were unrelated to the presenting pain (and thus truly incidental), their presence underscores the importance of documenting even benign IFs, as they could become symptomatic in the future.

Hepatobiliary incidental findings constituted the second most prevalent group. Hepatic IFs were found in 80 patients (7.5% of the total cohort), including fatty liver in 43 patients (4.0%) and hepatic cysts in 28 patients (2.6%). Nine patients (0.8%) had small hepatic hemangiomas. These liver findings were generally benign; for instance, none of the hepatic cysts were large (>4 cm) or symptomatic, and thus none required intervention [9]. Gallbladder-related IFs were identified in 16 patients (1.5%). This included 9 patients (0.8%) with asymptomatic gallstones and 7 patients (0.7%) with gallbladder adenomyomatosis. Gallbladder adenomyomatosis, although usually benign, can radiologically mimic malignancy or occasionally lead to symptoms [16]. Its incidental detection in our young cohort was deemed clinically significant enough to warrant periodic ultrasound surveillance to monitor for any changes, given that 0.7% of all patients had this finding.

Less common but noteworthy incidental findings in the pancreatic and adrenal systems were also observed. Two patients (0.2%) had pancreatic lesions detected incidentally: one case of an intraductal papillary mucinous neoplasm (IPMN) of the pancreas and one pancreatic cyst. Both pancreatic lesions were clinically significant IFs given their potential for malignant transformation; we recommended specialist gastroenterology follow-up for these (each representing 0.1% of the total population). Adrenal gland lesions were found in four patients. Two patients (0.2%) had adrenal hyperplasia and two (0.2%) had adrenal adenomas incidentally noted on the APCT. One of the adrenal adenomas was larger than 2 cm, a size at which further endocrinologic evaluation is typically advised to rule out hormonal activity or malignancy. All adrenal lesions (0.4% of the cohort in total) were flagged for clinical follow-up; even in the absence of overt symptoms, incidental adrenal masses in young patients necessitate evaluation given their potential clinical implications.

Several gastrointestinal tract-related IFs were detected as well. Colonic diverticulosis was seen in 8 patients (0.8%), an incidental finding given the young age of our cohort and one that generally required no acute intervention. Two patients (0.2%) had appendiceal mucoceles (mucus-filled appendiceal lesions) discovered incidentally on their CT. Both cases of appendiceal mucocele were considered clinically significant due to the risk of progression to malignancy (low-grade appendiceal mucinous neoplasm) [17]; consequently, both patients underwent surgical resection of the appendix [17]. Additionally, 2 patients (0.2%) were noted to have internal hemorrhoids on their scans—an incidental finding of minimal acute significance, documented for completeness. Miscellaneous abdominal findings included inguinal hernias in 3 patients (0.3%). These hernias were incidental in that they were unrelated to the abdominal pain presentation. All three were clinically apparent on exam as well, and surgical correction was performed, although the hernias were not the cause of the ED visits.

Incidental findings in non-abdominal structures captured on the APCT were also recorded. Musculoskeletal IFs were seen in 18 patients (1.7%). These included degenerative changes such as spinal spondylosis in 8 patients (0.8%), sacroiliitis in 4 (0.4%), scoliosis in 4 (0.4%), and a congenital variant (lumbarization of S1) in 2 patients (0.2%). While these bone and joint findings were unrelated to the acute abdominal issues, they could have long-term implications for a soldier’s physical fitness and were thus noted for outpatient follow-up as needed. Pulmonary incidental findings were observed in 8 patients (0.8%), specifically small pulmonary nodules seen at the lung bases on the APCT. It should be noted that the APCT includes only a portion of the lungs; thus, these pulmonary nodules were incidentally visualized at the lung bases and this does not represent a systematic chest CT screening of the entire lungs. All patients with incidental lung nodules were advised to undergo appropriate follow-up (such as dedicated chest imaging in a few months) to ensure these nodules are stable and likely benign findings.

Co-occurrence patterns among IF categories are visualized in a heatmap (Figure 2). The diagonal shows counts for each single IF, and off-diagonal cells show pairwise co-occurrence within the same APCT; zeros indicate no observed co-occurrence. Renal cysts most often co-occurred with hepatic cysts and fatty liver, whereas overlaps involving pulmonary nodules were infrequent. This display is descriptive and should be interpreted with caution for very small cells (e.g., n < 5).

#### 3.1.2. Clinically Significant Incidental Findings

Out of the 218 patients with IFs, only a subset had findings that required clinical intervention or specialized follow-up. Table 3 summarizes the clinically significant IFs and the recommended management for each. Renal lesions with malignant potential or significant pathology were the most prominent: the 3 Bosniak IIF renal cysts (0.3% of all patients) were scheduled for interval imaging follow-up due to their indeterminate nature, and the single cases of ADPKD and MCKD (0.1% each) were referred to nephrology/urology specialists for comprehensive management of these inherited renal conditions. The 7 cases of gallbladder adenomyomatosis (0.7%) were advised to undergo periodic ultrasound surveillance, given the need to distinguish them from more serious pathology over time and monitor for any changes. Incidental pancreatic lesions (the 1 IPMN and 1 pancreatic cyst, 0.1% each) were referred to gastroenterologists for further evaluation; in young patients, these lesions necessitate careful follow-up (and in the case of IPMN, consideration of surgical resection depending on size and features) [11,18]. Adrenal incidentalomas—2 patients with adenomas (0.2%)—were recommended for endocrine evaluation, especially the one >2 cm, to test for hormone secretion and assess if surgical removal might be warranted; the 2 cases of adrenal hyperplasia (0.2%) were also noted for endocrine follow-up to rule out subclinical adrenal dysfunction [19]. Both patients with appendiceal mucocele (0.2%) proceeded to surgical treatment (appendectomy) soon after detection, given the risk of neoplastic transformation. The 3 incidental inguinal hernias (0.3%) were electively repaired by surgeons (these were clinically evident and easily addressed). Finally, all 8 patients with incidental pulmonary nodules (0.8%) were scheduled for follow-up chest imaging (e.g., a dedicated chest CT or serial radiographs per Fleischner Society guidelines), to ensure that these nodules are stable and likely benign findings [12].

#### 3.1.3. Co-Occurrence of Incidental Findings (Cluster Analysis)

Descriptive co-occurrence patterns are shown in Figure 2; Our exploratory cluster analysis identified several FDR-significant co-occurrences of IFs (Table 4). Among patients with a renal cyst, 27.3% also had a hepatic cyst (*p* = 0.008, *q* = 0.016), higher than among those without renal cysts; likewise, 22.7% of patients with renal cysts had concomitant fatty liver (*p* = 0.045, *q* = 0.045). Hepatic cysts were also linked to gallbladder adenomyomatosis: 17.9% of patients with a hepatic cyst had adenomyomatosis (*p* = 0.032, *q* = 0.043). The apparent association between appendiceal mucocele and inguinal hernia was numerically large (50.0%), but it was based on only two mucocele cases (one co-occurrence) and should therefore be treated as hypothesis-generating given the very small numbers and the instability of estimates in this setting.

These relationships are visualized in Figure 3 as a network of significant co-occurrences: nodes represent IF categories, and edges connect pairs that co-occurred more often than expected. Edge thickness denotes association strength (e.g., odds ratio), and line emphasis reflects statistical support (*p*-value). Edges supported by very small cell counts (e.g., ≤2 cases) are labeled and should be interpreted with caution.

Taken together, these patterns suggest that some IFs cluster rather than occurring at random—potentially reflecting shared risk factors or anatomic/embryologic relationships. Clinically, they can prompt a more deliberate search for commonly co-occurring IFs during the same APCT and guide guideline-concordant follow-up, while avoiding over-interpretation of rare, small-cell signals.

## 4. Discussion

The 20.5% prevalence of incidental findings (IFs) in our study is somewhat lower than the 30–35% range reported in general civilian emergency populations [2,20]. This difference is likely attributable to our cohort’s unique demographics—young, healthy individuals screened via rigorous pre-enlistment examinations.

Although direct international comparisons with similar military cohorts are challenging due to a scarcity of research, an indirect comparison can be made using data from young civilian trauma patients. For instance, a prior study on trauma patients reported a 32.3% prevalence of IFs on abdominal CT, of which 7.2% required further investigation [20]. Notably, another study reported a 19.9% prevalence of IFs in trauma patients aged 40 and younger, a figure remarkably similar to the 20.5% found in our study [21].

This comparison highlights that our findings are significant, demonstrating that a substantial number of clinically meaningful lesions can be detected even in a presumably healthy young population. Therefore, the need for a systematic approach to manage and track IFs within the military healthcare system is underscored, particularly considering the unique environment of military service and the potential for discontinuity of care after discharge.

### 4.1. Clinical Implications of Incidental Findings

Despite the lower overall prevalence, our findings show that even in a highly screened, ostensibly healthy population, APCT can uncover clinically significant pathologies that warrant follow-up (e.g., ADPKD, IPMN, Bosniak IIF cysts). Although uncommon, such findings may carry important long-term implications for individual health and military readiness. These observations underscore the value of systematic, guideline-anchored interpretation and communication of IFs. Given the absence of longitudinal outcome data in this study, however, management implications should be viewed as supportive rather than confirmatory.

Hepatic and biliary IFs were among the more frequent categories. We observed fatty liver in 4.0% and hepatic cysts in 2.6% of all patients—percentages lower than some civilian reports but notable for this age group. Lifestyle patterns in military settings (e.g., high caloric intake with intermittent exercise) may contribute to these findings, although causality cannot be inferred from our data [22].

### 4.2. Co-Occurrence of Incidental Findings

Our cluster analysis was exploratory and should be considered hypothesis-generating. Several IFs tended to co-occur more often than expected—for example, renal and hepatic cysts—which may suggest shared predispositions (genetic or environmental) documented in prior literature [23]. The apparent association between appendiceal mucocele and inguinal hernia was numerically large, but it was based on only two mucocele cases (one co-occurrence) and is therefore highly sensitive to small-sample variation; this signal should be interpreted with caution and validated in larger cohorts. Practically, such patterns can encourage a deliberate search for commonly co-occurring IFs during the same APCT, while avoiding over-interpretation of rare, small-cell signals.

### 4.3. Military Healthcare Implications

Operationally, our results reinforce the need for structured management of IFs in resource-constrained military environments characterized by limited on-site subspecialty coverage and frequent unit transfers. Feasible steps include:A standardized IF section in radiology reports with guideline-linked next steps;EHR-embedded alerts (size/feature-triggered reminders with default due dates);A lightweight IF registry that persists across transfers to preserve follow-up tasks;Standardized handoff notes at discharge/transfer listing IFs and due dates;Periodic quality dashboards (completion rates, time-to-follow-up) [24].

Programs such as FIND have shown that systematic tracking and reminders can improve follow-up adherence for IFs [25]; adapting similar tools in the military health system could help ensure appropriate actions even when soldiers relocate.

### 4.4. External Validity

Generalizability is inherently limited by our young, male, active-duty cohort. Results should not be extrapolated to women, older adults, or civilian ED populations without caution. Moreover, system-level features of military care—frequent relocations, variable access to subspecialists, and mission-related interruptions—can affect the feasibility and timeliness of recommended follow-up. These differences may partly explain the lower IF prevalence we observed relative to mixed-age civilian cohorts and should be considered when interpreting our estimates.

### 4.5. Limitations

This retrospective, single-center case series has several inherent limitations. First, the retrospective design and convenience sampling introduce potential selection and information biases; reliance on existing ED documentation and archived imaging means some IFs or clinical details may have been missed or incompletely recorded. The high exclusion rate (34%)—largely related to missing or nonreviewable imaging and incomplete charts—may also bias prevalence estimates if excluded encounters differed systematically from those analyzed; characteristics of excluded cases were not consistently available, limiting our ability to quantify this risk. Second, we lacked longitudinal follow-up (e.g., completion of recommended surveillance, interval change, or downstream clinical consequences), which is particularly relevant in military settings where transfers and separation can lead to loss to follow-up; therefore, implications for management should be interpreted as hypothesis-supporting rather than confirmatory. Third, although we used independent double reading with consensus (and third-reader adjudication when needed) to improve detection, we did not compute formal inter-observer agreement statistics (e.g., Cohen’s κ), which may limit reproducibility. Fourth, “clinically significant” IFs were operationalized using contemporary consensus guidelines; because recommendations can vary across regions and have evolved over time, our mapping choices could have influenced categorization. Finally, exploratory co-occurrence/cluster signals were based on small cell counts (e.g., appendiceal mucocele with inguinal hernia) and the study was not powered to test rare associations; these findings should be considered hypothesis-generating and interpreted with caution. The homogenous, young, male military cohort further limits external validity to women, older adults, and civilian ED populations.

## 5. Conclusions

Incidental findings on APCT are common and clinically important even in a young, healthy military population. This study uncovers a significant and unaddressed health surveillance gap. The implementation of systematic, technology-driven protocols for reporting and tracking incidental findings is not merely a recommendation but a strategic priority to ensure the long-term health, readiness, and combat effectiveness of the armed forces.

## Figures and Tables

**Figure 1 healthcare-13-02736-f001:**
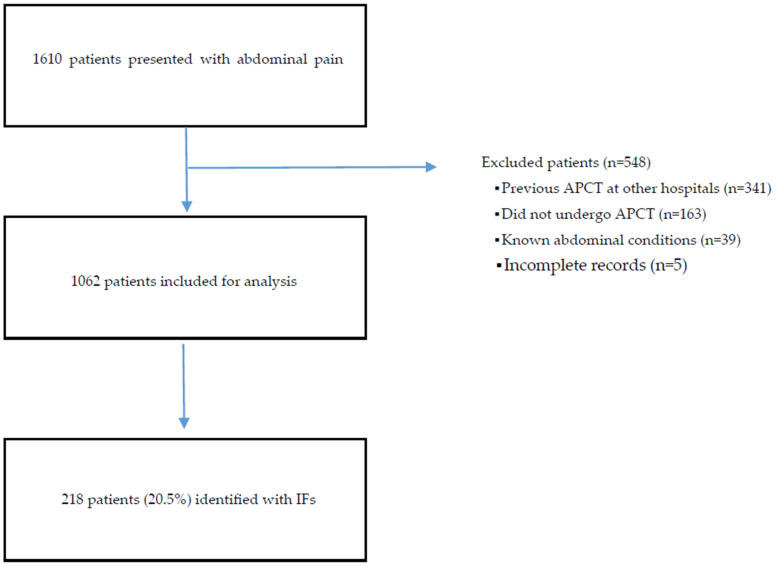
Study flow (STROBE). ED encounters screened; exclusions with reasons; final analytic cohort. Percentages are relative to the preceding box. ED = emergency department.

**Figure 2 healthcare-13-02736-f002:**
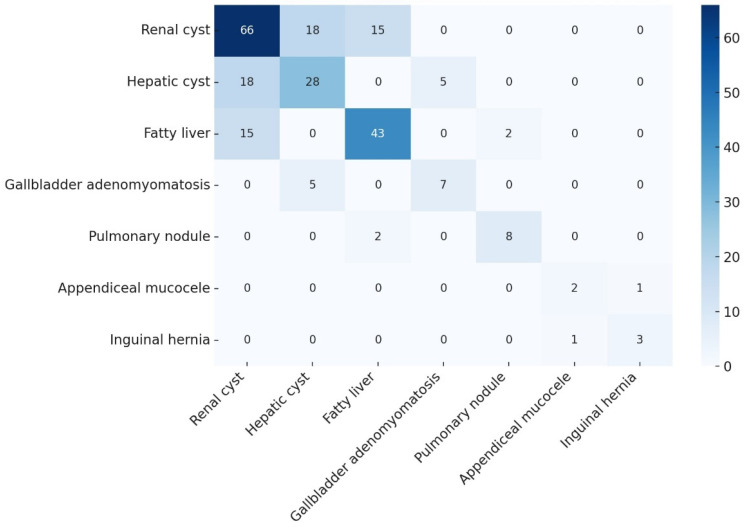
Heatmap of Co-occurrence Patterns among IFs. Each cell shows the number of patients (n) who had the IF on the row and the IF on the column in the same APCT. The diagonal displays the count of patients with each single IF. Color intensity encodes the magnitude of the count (see color bar); zeros indicate no observed co-occurrence. This heatmap is descriptive; statistical tests (e.g., Fisher’s exact test and corresponding *p*-values) are reported in the results section.

**Figure 3 healthcare-13-02736-f003:**
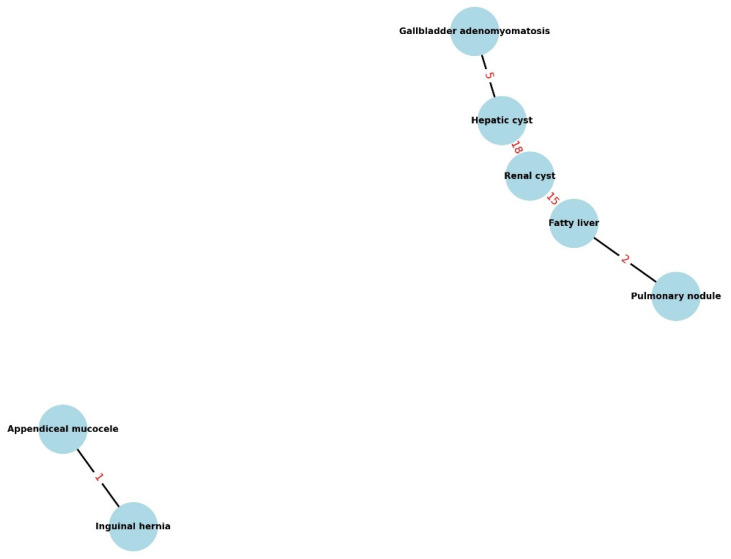
Network of IF co-occurrences: nodes = categories; edges = FDR-significant pairs (*q* < 0.05; see Table 4). Labels show patient counts (n); thicker lines ≈ stronger association (↑OR). Dashed edges indicate very small cells (≤2); colors are decorative only.

**Table 1 healthcare-13-02736-t001:** Demographic characteristics.

Characteristic *	With IFs (n = 218)	Without IFs (n = 844)	*p*-Value
Age (years), median (IQR)	21 (20–22)	21 (20–22)	0.14
Current smoker, n (%)	80 (36.7)	280 (33.2)	0.39
BMI (kg/m^2^), median (IQR]	24 (22–26)	23 (21–25)	0.08
Any previous medical history, n (%)	20 (9.2)	60 (7.1)	0.37

* Data are median (IQR) or n (%). Two-sided *p*-values from Mann–Whitney U or chi-square/Fisher’s exact tests.

**Table 2 healthcare-13-02736-t002:** Anatomical distribution of incidental findings on APCT scans.

Location	Incidental Finding	IF (n = 218), n (%)	All APCT Scans (n = 1062), n (%)
Liver and Spleen	Hemangioma	9 (4.1%)	9 (0.8%)
	Cyst	28 (12.8%)	28 (2.6%)
	Symptomatic or >4 cm cyst	0 (0%)	0 (0%)
	Polycystic liver disease (with multiple renal cysts)	1 (0.5%)	1 (0.1%)
	Fatty liver	43 (19.7%)	43 (4.0%)
	Hepatomegaly	5 (2.3%)	5 (0.5%)
	Splenomegaly	11 (5.0%)	11 (1.0%)
	Hepatosplenomegaly	2 (0.9%)	2 (0.2%)
Gallbladder	Asymptomatic stone	9 (4.1%)	9 (0.8%)
	Adenomyomatosis	7 (3.2%)	7 (0.7%)
Pancreas	Intraductal papillary mucinous neoplasm	1 (0.5%)	1 (0.1%)
	Cyst	1 (0.5%)	1 (0.1%)
Adrenal gland	Hyperplasia	2 (0.9%)	2 (0.2%)
	Adenoma	2 (0.9%)	2 (0.2%)
	Large adenoma (>2 cm)	1 (0.5%)	1 (0.1%)
Kidney	Cyst (any)	66 (30.3%)	66 (6.2%)
	– Single cyst	49 (22.5%)	49 (4.6%)
	>1 cm (subset of single cysts)	20 (9.2%)	20 (1.9%)
	– Multiple cysts	17 (7.8%)	17 (1.6%)
	• Largest > 1 cm	9 (4.1%)	9 (0.8%)
	– Cysts in both kidneys	7 (3.2%)	7 (0.7%)
	Bosniak classification:		
	• I	56 (25.7%)	56 (5.3%)
	• II	5 (2.3%)	5 (0.5%)
	• IIF	3 (1.4%)	3 (0.3%)
	• III/IV	0 (0%)	0 (0%)
	ADPKD (polycystic kidney disease)	1 (0.5%)	1 (0.1%)
	MCKD (medullary cystic kidney disease)	1 (0.5%)	1 (0.1%)
	Asymptomatic stone (kidney or ureter)	20 (9.2%)	20 (1.9%)
	Congenital anomalies:		
	• Horseshoe kidney	2 (0.9%)	2 (0.2%)
	• Incomplete kidney malrotation	1 (0.5%)	1 (0.1%)
	• Mal-rotated (downward) kidney	1 (0.5%)	1 (0.1%)
Gastrointestinal	Diverticulosis (colonic)	8 (3.7%)	8 (0.8%)
	Appendiceal mucocele	2 (0.9%)	2 (0.2%)
	Internal hemorrhoid	2 (0.9%)	2 (0.2%)
Perineal region	Inguinal hernia	3 (1.4%)	3 (0.3%)
	Prostatic cyst	5 (2.3%)	5 (0.5%)
	Urethral diverticulum	1 (0.5%)	1 (0.1%)
Spine/Pelvis	Scoliosis	4 (1.8%)	4 (0.4%)
(musculoskeletal)	Spondylosis (with spondylolisthesis)	8 (3.7%)	8 (0.8%)
	Lumbarization of S1	2 (0.9%)	2 (0.2%)
	Sacroiliitis	4 (1.8%)	4 (0.4%)
Lung	Pulmonary nodule	8 (3.7%)	8 (0.8%)
Other	Urachal remnant	2 (0.9%)	2 (0.2%)
	Abdominal wall hemangioma	1 (0.5%)	1 (0.1%)
	SMA with acute angle	1 (0.5%)	1 (0.1%)

**Table 3 healthcare-13-02736-t003:** Clinically significant incidental findings: implications and follow-up.

Incidental Finding	Patients (%)	Clinical Implication	Recommended Management
Gallbladder adenomyomatosis	7 (3.2%)	Potential malignancy risk	Periodic follow-up imaging
Bosniak IIF renal cyst	3 (1.4%)	Uncertain malignancy potential	Scheduled follow-up CT
Appendiceal mucocele	2 (0.9%)	Malignant potential	Surgical resection
ADPKD	1 (0.5%)	Renal dysfunction	Referral to specialist
Adrenal adenoma (>2 cm)	1 (0.5%)	Endocrine evaluation	Specialist follow-up

**Table 4 healthcare-13-02736-t004:** Cluster Analysis: Co-occurrence patterns among IFs.

Primary IF	Co-Occurring IF	Patients, n (%)	Odds Ratio (95% CI)	*p*-Value	*q*-Value
Renal cyst	Hepatic cyst	18 (27.3%)	2.3 (1.3–4.1)	0.008	0.016
Renal cyst	Fatty liver	15 (22.7%)	1.8 (1.0–3.3)	0.045	0.045
Hepatic cyst	Gallbladder adenomyomatosis	5 (17.9%)	3.1 (1.2–8.0)	0.032	0.043
Appendiceal mucocele	Inguinal hernia	1 (50.0%)	11.2 (1.5–85.0)	0.004	0.016

Benjamini–Hochberg FDR–adjusted *q*-values are reported; significance at *q* < 0.05.

## Data Availability

The data underlying this article cannot be shared publicly due to the privacy of study participants and the inclusion of sensitive military information. As the data were obtained from an Armed Forces hospital and include records of active-duty military personnel, they fall under military confidentiality regulations. Therefore, complete disclosure is not possible. However, data may be made available upon reasonable request to the corresponding author, subject to approval by the Ministry of National Defense of the Republic of Korea, and only after careful review and authorization.

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
