# Peer review of "Incidental Findings on Abdominopelvic CT in Young Korean Soldiers: Prevalence, Clinical Relevance, and Healthcare System Implications"

_healthcare, 2025, doi:10.3390/healthcare13212736_

Round 1

Reviewer 1 Report

Comments and Suggestions for Authors

Dear authors ,
The manuscript titled “Incidental Findings on Abdominopelvic CT in Young Korean Soldiers: Prevalence, Clinical Relevance, and Healthcare System Implications” addresses an important and underexplored topic. The focus on a young, ostensibly healthy military population adds novelty, and the study highlights gaps in incidental findings (IFs) follow-up within military healthcare systems. Overall, the paper is well-structured, and the results are clearly presented. However, a few areas need clarification and strengthening.

Major Comments:

  1. Study Limitations:

    The authors acknowledge retrospective design and single-center scope as limitations, but they should more explicitly discuss how the high exclusion rate (34%) may bias prevalence estimates.

    A lack of long-term follow-up is a major limitation, given the clinical implications of IFs—this point deserves stronger emphasis.
  2. Statistical Analysis: The cluster analysis is interesting, but the clinical relevance of findings based on very small numbers (e.g., appendiceal mucocele and hernia association) should be interpreted with much more caution. Please consider framing these as hypothesis-generating only.

  3. 3- Practical Implications: The discussion rightly highlights the importance of structured reporting and tracking systems. The authors could elaborate on feasible steps for implementation in resource-limited military settings. For example, are electronic alerts or integrated military health databases feasible in South Korea?
  4. Clarity and Consistency: Some sections (particularly Results) are very detailed. Consider condensing descriptive passages for readability. Minor inconsistencies in percentages and proportions should be double-checked for accuracy.

Minor Comments:

The abstract could better highlight the novelty of the study in contrast to civilian literature.

Please expand abbreviations (e.g., IFs) at first mention in both abstract and text.

Figure 2 (network graph) is useful but needs a clearer legend to guide interpretation.

Ensure references are consistently formatted according to journal style.

Overall Recommendation:
The paper is valuable and potentially suitable for publication after minor to moderate revisions. Strengthening the discussion of limitations, contextual comparisons, and practical implications will significantly improve the manuscript.

Author Response

We sincerely thank Reviewer #1 for the thorough and encouraging review. We are honored that you found our study original and methodologically sound, and we appreciate the opportunity to improve it with your guidance.

Reviewer 2 Report

Comments and Suggestions for Authors

The manuscript is clear, logically structured, and of potential interest to clinicians and policymakers. However, there are some areas to clarify, to improve the readability of your manuscript. 
in methods section, please provide more detail on how disagreements between radiologists were resolved (e.g., consensus vs. third reviewer); please explicity state whether inter-rater reliability (e.g., kappa statistics) was considered; clarify how “clinically significant” findings were defined, especially when guidelines differ across regions.

emphasize the limitations of small numbers (e.g., appendiceal mucocele + hernia) in both the Results and Discussion

in discussion please expand slightly on how the findings compare to large-scale civilian studies (e.g., whether the lower prevalence observed here is consistent with other young/healthy cohorts)

The high exclusion rate (34%) may introduce bias; consider acknowledging how excluded patients might differ systematically from included ones. Please, specify in limitations

Author Response

We are very grateful to Reviewer #2 for the encouraging and helpful comments. We have addressed each of your suggestions to improve the manuscript's clarity.

Reviewer 3 Report

Comments and Suggestions for Authors

I think that this manuscript addresses an important and underexplored issue: the prevalence and clinical implications of incidental findings (IFs) on abdominopelvic CT scans in young Korean soldiers. The study is original, methodologically solid, and provides relevant insights into the challenges of follow-up and continuity of care in the military healthcare system. The manuscript is clearly written, well-structured, and supported by appropriate statistical analysis. Nevertheless, several points need to be clarified, expanded, or revised to improve the clarity, impact, and scientific rigor of the work.

  1. The study population is highly specific (young, healthy, male Korean soldiers). While this is acknowledged in the limitations, it would strengthen the manuscript to further emphasize that results cannot be generalized to civilian populations, older adults, or female patients. Please expand the discussion on how this unique cohort affects external validity.

  2. The exclusion rate (34%) is high and could potentially bias the findings. Please provide more detail on the characteristics of excluded cases and discuss whether their exclusion may have systematically influenced the prevalence of incidental findings.

  3. The study reports incidental findings and recommends follow-up strategies, but it lacks longitudinal outcome data. Without confirming whether patients actually received follow-up or developed clinical consequences, the conclusions remain hypothetical. Please make this limitation more explicit and avoid over-stating clinical impact where direct evidence is absent.

  4. The exploratory cluster analysis is an interesting addition, but some associations are based on very small numbers (e.g., appendiceal mucocele and inguinal hernia, n=2). The manuscript should clearly label these findings as hypothesis-generating and interpret them with greater caution. Consider toning down the emphasis on such associations.

  5. Some references are outdated (e.g., Bosniak 2005). Please include more recent consensus updates and reviews (e.g., Bosniak classification 2019 revision). 

    • Expand the discussion to compare your prevalence rates with other international studies in young or military populations, not just Korean data. This would situate your findings in a broader context.
    • Expand the discussion to compare your prevalence rates with other international studies in young or military populations, not just Korean data. This would situate your findings in a broader context

  6.  There are several redundancies between the abstract, results, and discussion sections. Streamlining would improve readability and focus. The limitations section could be expanded to explicitly mention the lack of inter-observer agreement statistics (e.g., kappa values) between the two radiologists.

  7.  

    • Figures and Tables: Figures 1 and 2 are informative, but the legends could be expanded for clarity, especially for readers unfamiliar with network analysis.

    • Ensure that all abbreviations in tables are defined either in footnotes or in the text.

  8. Terminology

    • Standardize terminology (e.g., “emergency department” vs. “emergency room”).

    • Ensure consistent use of “incidental findings (IFs)” throughout the manuscript.

  9. Abstract

    • Consider rephrasing the conclusion of the abstract to more cautiously reflect the lack of outcome data. The current phrasing might give the impression that follow-up interventions were implemented and monitored.

Author Response

We thank Reviewer #3 for the positive feedback and for highlighting areas where we could add depth and clarity. Your comments have significantly improved the practical relevance of our work.

Round 2

Reviewer 3 Report

Comments and Suggestions for Authors

I have carefully reviewed the revised version of your manuscript and would like to commend you on the excellent work you have done to improve it. The paper is now much clearer, more balanced, and scientifically rigorous compared with the previous version. It is evident that you have addressed each reviewer’s comment thoughtfully and thoroughly, leading to a stronger manuscript both methodologically and editorially.

I particularly appreciate the expanded discussion of the study’s limitations and external validity. You have clearly articulated how the specificity of the military population affects generalizability and have maintained a properly cautious tone when discussing clinical implications. The issue of the high exclusion rate has also been handled transparently and described in sound methodological terms. The restructuring of the Discussion into well-defined subsections makes the narrative easier to follow and more accessible to readers.

The inclusion of updated references (such as the Bosniak 2019 revision, ESE/ENSAT 2023 guidelines, and the Fukuoka/Kyoto criteria) was an excellent choice, and the revised abstract now appropriately reflects the lack of longitudinal outcome data. I also appreciate that you have explicitly labeled small-sample co-occurrence findings as “hypothesis-generating,” demonstrating both scientific rigor and interpretative caution. The figures and tables are clearer and well-captioned, and the terminology is consistent throughout the text.

Overall, the manuscript is solid, well-written, and of genuine clinical and public health relevance, particularly in the context of military healthcare systems.

My only remaining suggestions are minor and editorial in nature: please specify the type of CT technology used (e.g., multidetector, with or without contrast) in the Methods section, and consider stating in the abstract’s opening line that this is a retrospective case series. Apart from these small details, the study appears fully ready for publication.

Congratulations again on the quality and clarity of your revision, and on the care with which you have addressed the previous review’s feedback.

Author Response

Response to the Reviewer

We sincerely thank the reviewer for the generous and encouraging assessment, and for the clear editorial suggestions. We have implemented both requested items as follows:

  1. Abstract

 Revised to explicitly state the study design:
Background: This retrospective case series examines incidental findings (IFs) detected on abdominopelvic CT (APCT) among young Korean soldiers presenting with acute abdominal pain.” (Abstract, first sentence)

  1. Methods

 Added a brief “CT acquisition and protocol” paragraph and aligned the eligibility text for consistency:

    • “All abdominopelvic CT examinations included in this analysis were performed on multidetector CT scanners (Canon computed tomography TSX-305A; Canon Medical Systems Corporation, Otawara, Japan) with intravenous iodinated contrast and portal venous–phase acquisition per institutional protocol.” (Methods §2.3)

We are grateful for the reviewer’s positive evaluation of our revisions (expanded limitations/external validity, updated references, clarified small-sample signals, improved figures/tables, and terminology consistency). We believe these additional edits further enhance clarity and transparency.

Sincerely,
Donghyoun Lee, MD (corresponding author)
on behalf of all co-authors